# A network model of affective odor perception

**Yingxuan Liu[1], Alexander Toet[1]\*, Tanja Krone[2], Robin van Stokkum[2], Sophia Eijsman[1], Jan B. F. van Erp[1,3]**

**1** Perceptual and Cognitive Systems, TNO, Soesterberg, The Netherlands, **2** Risk Analysis for Products in Development RAPID, TNO, Zeist, The Netherlands, **3** Research Group Human Media Interaction, University of Twente, Enschede, The Netherlands

\* lex.toet@tno.nl

**Data Availability Statement:** A CSV file with all results and the R code used to construct the network models are available from the OSF Repository at https://osf.io/psvcy/ with DOI: 10.17605/OSF.IO/PSVCY.

## Abstract

The affective appraisal of odors is known to depend on their intensity (I), familiarity (F), detection threshold (T), and on the baseline affective state of the observer. However, the exact nature of these relations is still largely unknown. We therefore performed an observer experiment in which participants (N = 52) smelled 40 different odors (varying widely in hedonic valence) and reported the intensity, familiarity and their affective appraisal (valence and arousal: V and A) for each odor. Also, we measured the baseline affective state (valence and arousal: BV and BA) and odor detection threshold of the participants. Analyzing the results for pleasant and unpleasant odors separately, we obtained two models through network analysis. Several relations that have previously been reported in the literature also emerge in both models (the relations between F and I, F and V, I and A; I and V, BV and T). However, there are also relations that do not emerge (between BA and V, BV and I, and T and I) or that appear with a different polarity (the relation between F and A for pleasant odors). Intensity (I) has the largest impact on the affective appraisal of unpleasant odors, while F significantly contributes to the appraisal of pleasant odors. T is only affected by BV and has no effect on other variables. This study is a first step towards an integral study of the affective appraisal of odors through network analysis. Future studies should also include other factors that are known to influence odor appraisal, such as age, gender, personality, and culture.

## 1 Introduction

### 1.1 The affective appraisal of odors

Odors can effectively elicit affective responses [1–5], probably due to the high degree of overlap and connectivity between the neural systems mediating olfaction and emotion [6–10]. These affective responses mediate our perception of environmental input and can adapt our output, thus enabling us to respond in an appropriate way [11]. The affective response to odors is typically characterized by its valence (pleasantness or hedonic tone) and arousal [12, 13], while both dimensions are mediated by different neural substrates [14]. Brain imaging studies show that unpleasant and pleasant odors also activate different brain areas [8, 15–19] in asymmetric ways [17, 20]. Unpleasant odors are processed faster than pleasant ones [17, 21–23], eliciting

**Funding:** The authors received no specific funding for this work.

**Competing interests:** The authors have declared that no competing interests exist.

specific patterns of autonomic [24, 25] and olfactomotor responses [26, 27] and specific neural activation [14, 16, 18, 20, 28–30]. Also, unpleasant odors are also less prone to top-down influences such as priming [31], verbal context [32] and odor knowledge [33].

Pleasant odors positively affect mood and decrease arousal, while unpleasant odors have the opposite effect [34]. It has been observed that unpleasant odors increase skin conductance, heart rate [35–37] and the startle reflex [38–40] while pleasant odors decrease these parameters. As a result, odors can effectively be used to induce various emotional states [2, 41–43] and desired behaviors [11]. In real-life settings, odors have for instance effectively been deployed to reduce patient stress in healthcare environments [44–46], to influence shopping behavior in retail environments [47, 48] and to influence littering behavior in public environments [49]. Because the principal distinctive properties of food flavors are provided by olfaction rather than by taste cues [50], our culinary preferences are also to a large extent based on the affective appraisal of food odors. However, despite the important role of affect in olfaction, it is still largely unknown how affective appraisal and olfactory perception interact and converge in everyday life [9].

## 1.2 Factors related to the affective appraisal of odors

Factors that are known to be related to the affective appraisal of odors include odor sensitivity, odor intensity, odor familiarity (the feeling that an odor is known or has been perceived before: [33]) and core affective state [25, 51–54]. Previous studies only investigated the correlations between specific subsets of these factors. As a result, the extent to which individual differences in these factors and their interrelations differentially influence the affective response of people to specific odors is still largely unknown [2]. In this section we will first present the available evidence for the mediating effects of sensitivity, familiarity, and core or baseline affective state on affective odor appraisal. Fig 1 represents the known relations between these

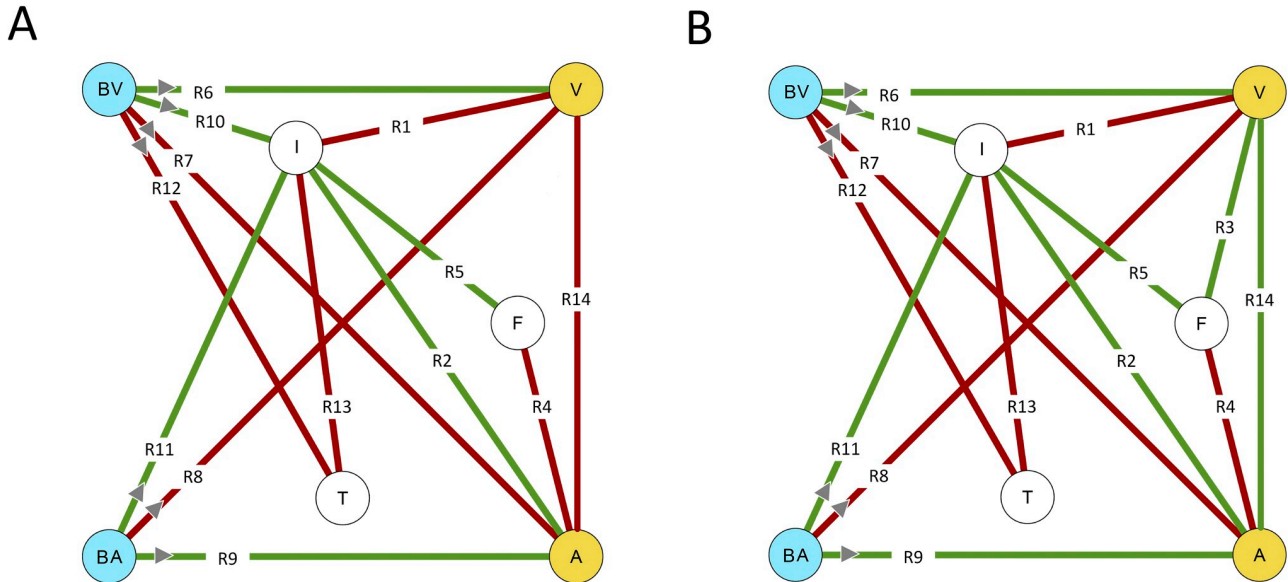

**Fig 1. Hypothetical odor-evoked affect (HOEA) network models for (a) pleasant (PHOEA) and (b) unpleasant (UHOEA) odors.** These networks represent the relations that have been reported in the literature between different factors influencing the affective appraisal of odors. Yellow nodes: the valence (V) and arousal (A) components of the affective odor appraisal. Blue nodes: the observer's baseline valence (BV) and arousal (BA) values. I: odor intensity. T: odor detection threshold. F: odor familiarity. Edge color represents the polarity of the partial correlations green = positive, red = negative, grey = positive for pleasant odors, negative for unpleasant odors. The edge labels serve to identify the relations for discussion in the text.

different factors as hypothetical graphical network models for affective odor appraisal (hypothetical odor evoked affect or HOEA model). Given the aforementioned evidence for the existence of different processing channels for unpleasant and pleasant odors, we will distinguish between an unpleasant (UHOEA: Fig 1a) and a pleasant (PHOEA: Fig 1b) model. In the next sections, we will refer to the relations between the variables in both HOEA models (indicated by R1-R15 in Fig 1) to facilitate the discussion.

**1.2.1 Intensity.** Odor intensity is generally negatively correlated with valence (R1): the more intense being the more unpleasant [51, 55]. However, intensity and valence interact in complex ways [55–58], involving both innately tuned and learned components. The polarity of the effect may also depend on the nature of the stimulus and on the perceiver's personal characteristics [52, 56, 57, 59, 60]. As a result, several exceptions to R1 have been observed, with some odors showing a positive correlation between valence and intensity, some a negative, and others an inverted U-shape or even an absence of correlation [55–57, 59]. Experience and learning significantly determine odor valence [61, 62]. Odor knowledge (identification) significantly enhances ratings of intensity, pleasantness and familiarity [63]. There are also indications that individuals with high detection thresholds may show a positive correlation of odor intensity with valence [56], although the evidence for this assumption is weak. Odor intensity is typically strongly positively correlated with subjective and autonomic indices of arousal (R2; [12]), independent of odor valence [14, 56, 57].

**1.2.2 Familiarity.** Familiarity is implicitly linked to the affective appreciation of our environment rather than to explicit source recognition [64]. Olfaction appears to serve novelty or change detection (possibly mediated by the amygdala: [9, 65, 66]), directing our attention to odors that are either unknown (not experienced before: categorical novelty) or do not fit our expectation or previous experience of a given situation (contextual novelty or misfit; see [64]).

Both direct and indirect effects of odor familiarity on affective odor appraisal have been reported in the literature.

The relation between familiarity and odor valence appears to be asymmetrical. For pleasant odors (Fig 1b), familiarity and odor valence are typically positively correlated: the more familiar an odor, the more pleasant it is judged (R3 in Fig 1b; e.g.: [12, 51–53, 63, 67–75]). For unpleasant odors, no consistent relation has been found (R3 is absent in Fig 1a) [25, 33, 51]. This finding agrees with the idea that different evaluative channels are involved in the processing of negatively and positively valenced stimuli [76]. In general, unpleasant odors are relatively less prone to top-down (cognitive) influences [31–33]. A negativity bias for unidentifiable pleasant odors may for instance reflect a behavioral system designed for self-protection that elicits a warning or avoidance response when confronted with a positive but unfamiliar (unknown or unexpected: [64]) odor that may represent a potential health threat [77].

Familiarity and odor evoked arousal are negatively correlated, independent of odor valence (R4): the more familiar pleasant (e.g., the comforting and relaxing smell of a familiar environment or the perfume of a loved one) and unpleasant (e.g., the smells of smoke or decay, signaling threat or danger) odors are, the less arousing they are judged [25].

It also appears that familiarity can indirectly influence the affective appraisal of odors by modulating their intensity: participants perceive familiar odors as more intense than unfamiliar odors (R5; [51, 52, 63, 70]), which may ultimately influence their valence (R1; [63]). Note that familiarity may differentially affect the affective appraisal of an odor depending on a person's history with it (e.g., due to a change in valence because of its contiguous presentation with a positive or negative event [78]).

**1.2.3 Affective state.** Both direct and indirect effects of affective state on the appraisal of affective stimuli have been reported in the literature.

Core or baseline affective state may have a direct impact on subsequent judgements through misattribution [79–82]. People are inclined to make cognitive appraisals of unrelated topics and objects reflecting their core affective state (R6-R9; [81, 83]). In particular, they tend to attribute residual arousal from prior experiences to external cues in subsequent situations (R9; [84]). Since this may also be the case for the affective appraisal of odors, we hypothesize that BV and V (R6) and BA and A (R9) may be positively correlated (a carry-over effect), while BV and A (R7) and BA and V (R8) may be negatively correlated (a contrast effect).

Core affective state can also indirectly influence the affective appraisal of odors. Affective state modulates chemosensory event-related potentials [54] and affects odor intensity (R10, R11): it has been observed that emotions enhance odor intensity [85, 86], independent of odor valence [60, 87]. Furthermore, emotional valence also modulates the odor detection threshold: a negative emotional state reduces olfactory sensitivity (R12; [54, 86]. This may in turn influence odor associated affect by modulating the odor intensity: people with elevated thresholds perceive odors as being less intense (R13; [88–91]). Although emotional arousal mediates the affective appraisal and intensity of odors [60, 86], there is currently no evidence that it directly modulates the odor detection threshold [86, 92].

## 1.3 Relation between valence and arousal

The general assumption of the independence between valence and arousal for the affective appraisal of affective stimuli has recently been questioned: although valence and arousal appear to be uncorrelated when valence is ambiguous, they tend to become correlated when valence is clear [93–95]. Hence, these dimensions may be correlated (R14) for the affective appraisal of odors with a clear valence. For a wide range of different affective stimuli it has been found that arousal generally increases (a) with increasing valence for positively valenced stimuli and (b) with decreasing valence for negatively valenced stimuli [93]. Therefore, we assume that both variables are positively correlated for pleasant odors (R14 is positive in Fig 1b) and negatively correlated for unpleasant odors in the HOEA model (R14 is negative in Fig 1a).

Note that the valence of odors may change due to learning effects. While affective odors appraisal appears to be partly innate [96–98], factors like the frequency and context of prior exposure, semantic knowledge, and cultural background can cause significant variations in hedonic perception between individuals and over the course of the human life-span [99]. For instance, odors that are initially perceived as neutral or positive may acquire a negative connotation and signal threat after they have been experienced in the context of negative life events [100].

## 1.4 Current study

The goal of this study was to explore the potential relations between the different variables in in our literature-based HOEA model (Fig 1). Since previous studies investigated these variables individually, there is currently no integral model for their interrelations. To fill this gap, we performed an observer experiment in which participants reported the valence and arousal, intensity and familiarity for a range of different odors, varying widely in hedonic valence. In addition, we measured the participants' baseline affective state and detection threshold. We explored the relations between baseline affective state, odor familiarity, odor intensity and odor detection threshold, and their impact on affective odor appraisal through probabilistic network analysis [101–104]. Network analysis is a data-driven exploratory approach to modelling, allowing model structure to spontaneously emerge from the statistical relationships among indicators, thereby eliminating the need to specify an a-priori model. Network analysis focusses on the direct relations between observed variables. Hence, network analysis and

visualization can yield new insights into the relations between variables. In psychology, network analysis has recently become a popular alternative for latent variable modelling in exploratory studies of human behavior [101, 103, 105–111]. Psychological networks consist of nodes representing observed variables (e.g., questionnaire items), connected by edges representing the statistical relationships between the variables (their pairwise interactions; [112]). Network analysis typically involves the following three steps [112]: (1) network estimation, (2) network analysis, and (3) network comparison.

In the rest of this paper we first present the methods, materials and techniques used in this study. Then we present the results and compare the network models that were estimated from our present results to the HOEA model that was based on findings from the literature. Finally, we discuss the implications of the current findings and the limitations of this study.

## 2 Methods

### 2.1 Participants

To conduct a power analysis (determine the adequate sample size) an expectation of the effect size is required. The network equivalent is an expected (weighted) network structure [112]. However, since this is the first study of its kind, no previous similar networks were available. Sample size was therefore determined from a general rule of thumb suggested in the literature; namely, three individuals per parameter [112]. Since the HOEA network has 14 edges, this means that this study required a minimal group size of 42 participants to meet this "rule of thumb".

A total of 56 students (32 females, and 24 males, mean age = 24.3 years, SD = 4.6) from Utrecht University (Utrecht, the Netherlands) participated in this experiment. Participants were recruited through postings on social media and direct messaging. The exclusion criteria were age (younger than 18 years and older than 60 years), olfactory deficiencies (e.g., diseases, having a cold, smoking or drinking alcohol) and pregnancy. Participants were asked not to wear perfume, use deodorant or wear scented clothing on the testing day. All participants signed an informed consent form. The experimental protocol was reviewed and approved by the TNO Internal Review Board (TNO, the Netherlands: reference 2019–024) and was in accordance with the Helsinki Declaration of 1975, as revised in 2013 [113]. After completing the study, participants were offered a small compensation (5 Euro or study credits) for their participation.

### 2.2 Stimuli

In this study we measured odor-evoked valence and arousal for 40 different odors (see Table 1), ranging from unpleasant and arousing (e.g., feces, fish), via pleasant and calming (e.g., clove, cinnamon) to pleasant and stimulating (e.g., peach, caramel). To obtain a stimulus set with valence values distributed across the entire scale range, we complemented the revised 32-item "Sniffin' Sticks" odor identification test, which contains neutral and pleasant smells (www.burghart-mt.de, see also: [114]), with eight additional odors that are typically perceived as unpleasant: burned wood, diesel fumes, dusty cave, metal, rhinoceros, tar (obtained from https://retroscent.com and indicated by the RS codes in Table 1) and with indole (unpleasant smell associated with feces) and wintergreen (typically perceived as less pleasant by Europeans: [34]; both obtained from www.hekserij.nl). The Sniffin' Sticks identification test consists of two sets (a blue capped set and a purple capped set) of 16 numbered felt pens each, with tips that are impregnated with 4 mL of fluid odor substance. This test is normally used to assess an individual's olfactory identification performance [115–117]. We prepared eight extra sticks by injecting 4 mL of the additional unpleasant odor substances in

**Table 1. Mean (SD) valence, arousal, familiarity and intensity ratings for all odors used as stimuli in this study.** The Sniffin' Sticks B and P codes refer to the Blue and Purple identification test sets (www.burghart-mt.de). The RS codes refer to the RetroScent product code (https://retroscent.com). Odors with a negative mean valence rating are printed in boldface.

| ID | Label | Code | Valence | Arousal | Familiarity | Intensity |
|----|-------|------|---------|---------|-------------|-----------|
| 1 | Anise | Sniffin' B15 | 0.34 (2.09) | 0.05 (2.02) | 72.63 (23.39) | 54.34 (20.91) |
| 2 | Apple | Sniffin' B11 | 0.84 (2.08) | 0.20 (1.98) | 55.07 (23.54) | 61.89 (19.19) |
| 3 | Banana | Sniffin' B5 | 1.47 (1.72) | 0.48 (2.24) | 77.25 (17.76) | 60.18 (20.14) |
| 4 | **Burned wood** | RS-420 | -1.54 (2.15) | 0.76 (2.09) | 49.86 (28.08) | 73.07 (21.13) |
| 5 | Caramel | Sniffin' P15 | 2.25 (1.39) | 1.11 (2.25) | 80.50 (13.62) | 58.75 (20.44) |
| 6 | Cinnamon | Sniffin' B3 | 0.98 (1.98) | 0.22 (2.40) | 61.88 (29.39) | 54.25 (23.22) |
| 7 | **Cloves** | Sniffin' B12 | -0.33 (2.30) | -0.02 (2.23) | 51.38 (29.16) | 62.88 (21.05) |
| 8 | Coconut | Sniffin' P9 | 1.94 (1.59) | 0.87 (2.10) | 77.32 (19.03) | 56.16 (19.42) |
| 9 | Coffee | Sniffin' B10 | 0.77 (2.26) | 0.33 (2.23) | 72.52 (27.13) | 58.75 (22.12) |
| 10 | Coke | Sniffin' P2 | 0.37 (1.77) | -0.48 (1.86) | 48.11 (25.65) | 49.84 (20.35) |
| 11 | **Diesel fumes** | RS-423 | -1.76 (1.55) | -0.04 (2.22) | 48.84 (23.97) | 60.86 (21.47) |
| 12 | **Dusty cave** | RS-425 | -0.20 (1.82) | -0.76 (1.89) | 36.59 (22.94) | 43.00 (25.00) |
| 13 | Eucalyptus | Sniffin' P7 | 0.66 (2.09) | 0.21 (2.11) | 73.55 (22.77) | 70.36 (19.14) |
| 14 | **Fish** | Sniffin' B16 | -2.07 (1.74) | 0.84 (2.34) | 63.16 (27.69) | 71.39 (26.26) |
| 15 | **Garlic** | Sniffin' B9 | -1.18 (2.20) | 0.79 (2.32) | 68.30 (26.61) | 77.21 (18.14) |
| 16 | Ginger | Sniffin' P8 | 0.01 (1.91) | -0.52 (1.87) | 47.38 (24.41) | 55.36 (20.81) |
| 17 | Grapefruit | Sniffin' P4 | 0.83 (1.91) | 0.12 (2.11) | 53.20 (24.58) | 55.48 (19.65) |
| 18 | Grass | Sniffin' P5 | 0.07 (2.04) | -0.14 (2.03) | 67.43 (22.45) | 63.93 (21.36) |
| 19 | **Feces** | Indole | -1.97 (1.86) | 0.53 (2.16) | 40.61 (24.86) | 65.13 (22.17) |
| 20 | Lavender | Sniffin' P10 | 0.93 (1.88) | -0.08 (2.20) | 66.91 (25.23) | 56.64 (18.45) |
| 21 | **Leather** | Sniffin' B2 | -0.57 (1.91) | -0.71 (1.54) | 41.52 (25.13) | 46.16 (20.95) |
| 22 | Lemon | Sniffin' B6 | 1.49 (1.73) | 0.48 (2.12) | 60.14 (25.38) | 53.70 (21.15) |
| 23 | Lilac | Sniffin' P3 | 0.92 (2.02) | -0.32 (2.05) | 68.00 (22.94) | 59.23 (20.04) |
| 24 | Liquorice | Sniffin' B7 | 0.92 (2.13) | 0.00 (1.93) | 78.23 (24.00) | 57.04 (19.83) |
| 25 | Melon | Sniffin' P11 | 1.58 (1.68) | -0.12 (2.38) | 66.66 (21.52) | 54.64 (22.07) |
| 26 | **Metal** | RS-426 | -0.71 (1.75) | -0.59 (1.82) | 36.95 (25.43) | 50.09 (22.73) |
| 27 | **Mushroom** | Sniffin' P13 | -1.28 (2.10) | 0.51 (2.12) | 45.13 (29.10) | 66.30 (18.68) |
| 28 | **Onion** | Sniffin' P16 | -1.73 (2.03) | 1.00 (2.12) | 55.66 (28.30) | 69.20 (21.28) |
| 29 | Orange | Sniffin' B1 | 2.04 (1.30) | 0.57 (2.23) | 76.25 (19.69) | 54.88 (24.05) |
| 30 | Peach | Sniffin' P12 | 2.51 (1.67) | 1.55 (2.07) | 79.70 (19.67) | 61.84 (22.93) |
| 31 | Pear | Sniffin' P1 | 1.14 (1.74) | 0.02 (2.11) | 59.00 (22.74) | 57.23 (19.97) |
| 32 | Peppermint | Sniffin' B4 | 1.63 (1.80) | 0.57 (2.02) | 84.84 (16.30) | 64.80 (23.05) |
| 33 | Pineapple | Sniffin' B13 | 1.21 (2.07) | 0.22 (2.17) | 58.93 (26.68) | 64.77 (18.26) |
| 34 | Raspberry | Sniffin' P6 | 2.20 (1.23) | 0.57 (2.18) | 66.54 (18.73) | 54.77 (21.88) |
| 35 | **Rhinoceros** | RS-424 | -2.10 (1.48) | 0.46 (2.31) | 49.34 (25.58) | 70.84 (20.02) |
| 36 | Rose | Sniffin' B14 | 1.43 (1.77) | -0.09 (2.13) | 72.77 (16.78) | 61.23 (21.94) |
| 37 | **Smoked meat** | Sniffin' P14 | -0.92 (1.82) | -0.57 (1.92) | 44.16 (26.65) | 58.54 (21.82) |
| 38 | **Tar** | RS-401 | -1.23 (2.20) | 0.63 (2.07) | 52.71 (26.88) | 72.59 (20.06) |
| 39 | **Turpentine** | Sniffin' B8 | -0.87 (1.94) | -0.55 (2.01) | 46.91 (26.11) | 57.77 (18.90) |
| 40 | **Wintergreen** | Gaultheria oil | -0.99 (2.19) | 0.02 (2.02) | 56.48 (29.89) | 69.30 (21.12) |

empty Sniffin' Sticks. Hence, our total stimulus set consisted of 40 sticks pens, numbered from 1 to 40 (see Table 1). Since extreme differences in intensity may confound the affective ratings because of the inverse valence-intensity relation, a panel consisting of three of the authors (SE, YL, AT) verified that the set of odor samples did not contain any outliers in intensity, prior to the experiments. To ensure the compatibility between the samples, we

adopted the criterion set by the developers of the Sniffin' Sticks that all intensities should be within about 25% of the mean intensity [116]. The same set of sticks was used during the entire experiment. All samples were prepared in compliance with the safety Standards of the International Fragrance Association [118].

### 2.3 Measures

**2.3.1 Odor detection threshold.** Odor detection thresholds were measured using the standard "Sniffin' Sticks" odor threshold test (www.burghart-mt.de) in combination with a single-staircase, triple-forced-choice procedure [117]. The test comprises 16 triplets of pens (total of 48 pens). The three pens in each triplet are distinguished by the color of their cap (red, green and blue). Red pens are impregnated with phenylethylalcohol (PEA) diluted in a solvent according to decreasing concentrations. Blue and green pens are only impregnated with solvent. During the test, participants were blindfolded with a sleep mask to prevent them from recognizing the odorant-containing pens. For odor presentation, a pen's cap was removed by the experimenter for about 3 s and the pen's tip was placed approximately 2 cm below both nostrils of the participant. The three pens of a triplet (two containing only the solvent and one containing also the odorant) were presented in a randomized order. Participants were asked to detect the odor-containing pen in each triplet (forced choice). Triplets were presented at intervals of approximately 20 s. Reversal of the staircase toward lower concentrations was triggered either when the odor was correctly detected in two successive trials or toward higher concentrations when the odor was not detected in a trial. The total number of reversals was seven, and the threshold (T) was defined as the arithmetic mean of the last four staircase reversals. There was no absolute number of correct responses required. The subjects' scores ranged between 1 (lowest sensitivity or highest threshold: no odor detected) and 16 (highest sensitivity or lowest threshold).

**2.3.2 Valence and arousal.** The graphical EmojiGrid affective self-reporting tool (Fig 2; [119]) was used to measure subjective valence and arousal. The EmojiGrid is a Cartesian axes system similar to the Affect Grid [120], but the verbal labels on the midpoints and endpoints of the axes are replaced with emoji showing facial expressions. Also, additional emoji are inserted between the midpoints and the endpoints of each axis (resulting in five emoji on each side of the grid), and one (neutral) emoji is placed in the center of the grid, resulting in a total of 17 emoji on the grid. The central emoji with a neutral expression serves as a baseline or anchor point. The facial expressions of the emoji vary from disliking (unpleasant) via neutral to liking (pleasant) along the horizontal (valence) axis, and gradually increase in intensity along the vertical (arousal) axis. The facial expressions are defined by the eyebrows, eyes and mouth configuration of the face, and are inspired by the Facial Action Coding System [121]. The arousal dimension is represented by the opening of the mouth and the shape of the eyes, while the valence dimension is represented by the concavity of the mouth, the orientation and curvature of the eyebrows, and the vertical position of these features in the face area (representing a slightly downward looking face for lower arousal values and a slightly upward looking face for higher valence values). Users respond by clicking on a point inside the grid that best represents their affective appraisal of the stimulus.

At the start of the experiment participants first rated their baseline affective state on the EmojiGrid. In the rest of the experiment they used the EmojiGrid to rate their affective appraisal of the 40 different odor stimuli. All ratings were scaled to a range between -4 and 4.

**2.3.3 Odor intensity and familiarity.** Familiarity (F) and intensity (I) of each odor were measured with two single-item questions: "*How intense do you perceive the scent?*" and "*How familiar are you with the scent?*". Participants rated these items by dragging a slider under each

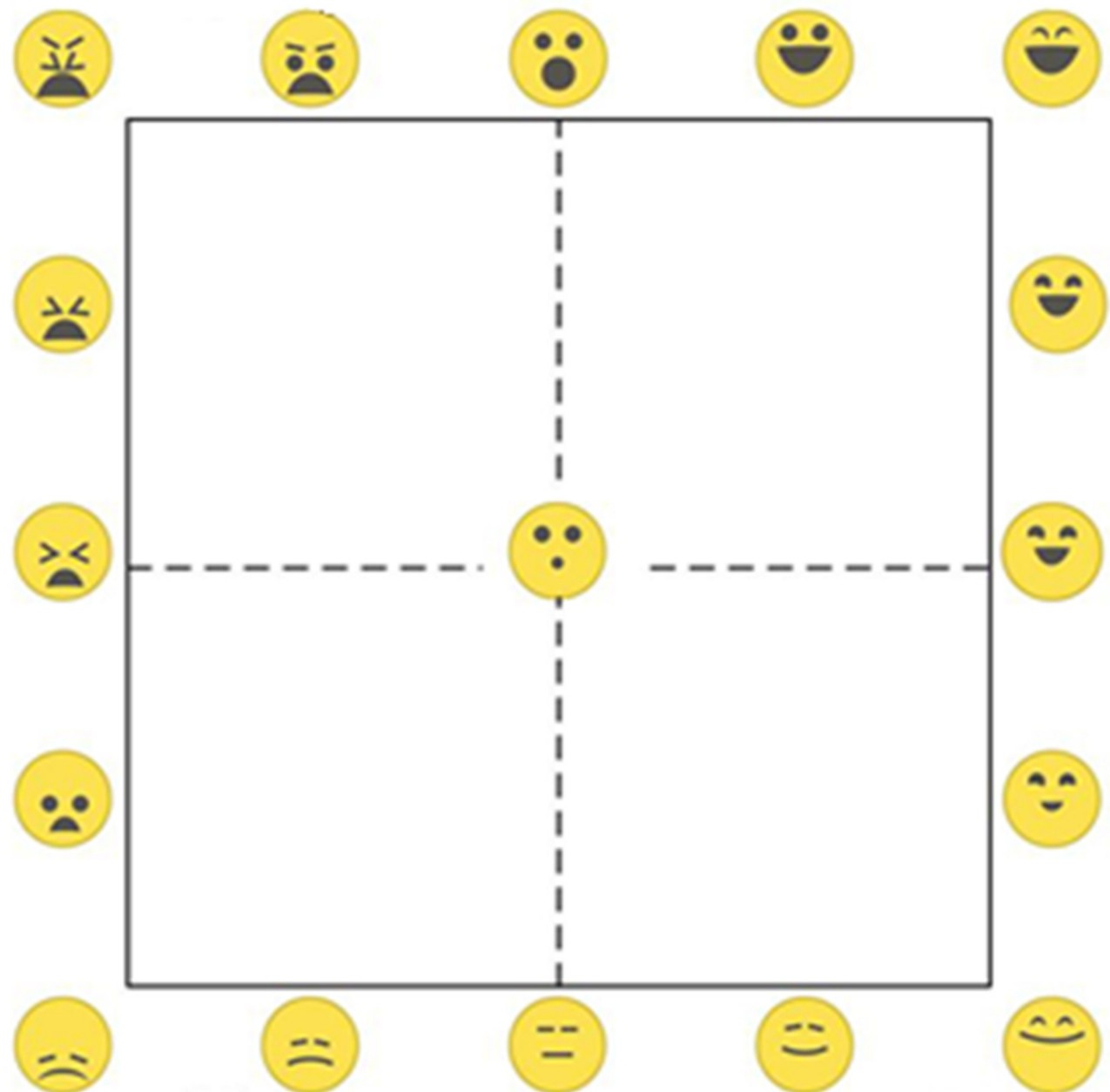

**Fig 2. The EmojiGrid: An emoji labeled Affect Grid for the measurement of odor-related affective associations.** The facial expressions of the emoji vary from disliking unpleasant via neutral to liking pleasant along the horizontal valence axis, and gradually increase in intensity along the vertical arousal axis.

question to a value between 0 and 100. The slider defaulted at 50. Participants could see a tool-tip with the current slider value while rating.

## 2.4 Procedure

The tests were performed in a quiet, well-ventilated room to avoid the presence of any residual odors. The experimenter wore odorless cotton gloves during the entire experiment. A

| Introduction | Detection threshold | Appraisal test |
|---|---|---|
| • Instructions<br>• Informed consent<br>• Demographics<br>• EmojiGrid introduction<br>• Blindfold application | • Identifying odor containing pen from successively presented triplets<br>• Blindfold removal | • Rating 40 scent pens :<br> • Valence and arousal<br> • Intensity<br> • Familiarity |

$T_0$ $T_0 + 10$ $T_0 + 20$ $T_0 + 60$

**Fig 3. Timeline of the events in the experimental procedure.**

computer was used to register all responses and to suggest a random stimulus presentation order to the experimenter.

Fig 3 shows the timeline of the events in the experimental procedure.

Upon their arrival at the laboratory, the participants were welcomed by the experimenter and received a verbal introduction and instructions. Then, they read and signed an informed consent. Next, they filled in their nationality, age and gender. Then the EmojiGrid was presented on a computer screen and the participants were asked to study it carefully. They were informed that they could respond by clicking on a point inside the grid that best represented their emotional state.

The experiment consisted of two parts. In the first part the odor detection threshold of the participants was determined. In the second part the participants rated the intensity, familiarity and the subjective valence and arousal for each of the 40 different odors. Before starting the odor measurements, the participants first rated their momentary affective state (valence and arousal) using the computer-based EmojiGrid. Then they were blindfolded, and the odor detection threshold test started. After finishing the threshold test, the participants took off their blindfolds and the experimenter started the odor appraisal test. The participants were explicitly asked not to attempt to identify the smells since knowledge of odor sources may influence their valence, intensity and familiarity [33, 51, 63, 73, 122]. During the experiment, the experimenter presented each of the 40 scent pens once (after removing the cap of the pen) for about 5 seconds at a distance of about 2 cm from the edge of both nostrils of the participant. The presentation order was randomized over the participants. The participants sniffed following a brief verbal command (natural sniffing is known to provide optimal odor perception: [123]). Immediately after sniffing the pen was removed (and its cap replaced by the experimenter), and the participants were given at least 30 s to smell fresh air (to reduce potential effects of olfactory adaptation and habituation: [124]). During this interval, participants rated their affective appraisal (valence and arousal), intensity and familiarity of the smell. The entire experiment lasted about an hour.

## 2.5 Data analysis

**2.5.1 General statistics.** IBM SPSS Statistics 25 (www.ibm.com) was used to inspect the data for outliers (standardizing all ratings of intensity, familiarity, valence and arousal for each odor and for each participant) and to compute the mean values for valence and arousal for each odor over all participants.

Matlab 2019a (www.mathworks.com) was used to investigate the relation between the (mean) valence and arousal ratings and plot the data. The Curve Fitting Toolbox (version 3.5.7) in Matlab was used to compute a least-squares fit of a quadratic function to the data points. Based on this analysis (the mean valence ratings) the odors in the stimulus set were classified as either positive or negative.

All further data analysis was done in R version 3.6.0 (R Core Team, www.r-project.org) in R Studio 1.2.1335 (www.rstudio.com). The exact version numbers of all R packages used are documented in the S1 Data.

In all statistical analyses, a probability level of p < .05 was considered as statistically significant. To attenuate interindividual variance (as this is not the main interest of this paper) while retaining within-subject variance, we converted the valence, arousal, intensity and familiarity scores per individual and per odor valence set (pleasant/unpleasant) to z-scores [125]. Participants with standardized values exceeding two standard deviations from the mean were considered as outliers.

**2.5.2 Network estimation.** The most popular method to estimate network models for continuous and normally distributed data is the Gaussian Graphical Model (GGM: [126]). The GGM estimates a network of regularized partial correlations, thereby controlling for spurious relationships. When continuous data are not normally distributed, a transformation should be applied (e.g., a nonparanormal transformation; [127], see also [104]) to Gaussianize the input before estimating the GGM. In the resulting network, two connected variables are dependent after controlling for all other variables in the dataset. Thus, an edge connecting two nodes represents their conditional dependence given all other nodes. The absence of an edge between two nodes indicates that they are conditionally independent given all other nodes. The GGM has extensively been applied to psychological data [106, 107, 110, 128].

In this study we constructed GGMs to investigate the relations between affective state, odor sensitivity, odor intensity, odor familiarity and the affective appraisal of (positive and negative) odors. We used the nonparanormal transformation *huge.npn* from the *huge* R package [129] to normalize the data. Note that partial correlations can differ from zero due to sampling variation and may therefore represent false relations [130]. We therefore regularized our models with the graphical LASSO (Least Absolute Shrinkage and Selection Operator: [131]) algorithm, using the R packages *glasso* [131] and *qgraph* [132]. This procedure eliminates weak edges and returns a sparse network by driving low values of partial correlations to zero [104]. A sparse network is a parsimonious one that best accounts for the covariance among nodes while minimizing the number of edges. The LASSO algorithm first generates 1000 different network models with different degrees of sparsity (ranging from fully connected to fully disconnected), for 1000 different values of the tuning parameter λ that controls the level of sparsity [104, 133]. Then, it selects the model with the minimal EBIC (Extended Bayesian Information Criterion: [134]) value, given a value of the hyperparameter γ (which controls the trade-off between including potentially true edges and eliminating potentially false edges: [112]). The hyperparameter γ is usually set between zero and 0.5 [112]. As the value of γ approaches 0.5, the EBIC will favor a simpler network with fewer edges. In this study we set γ to its recommended default value of 0.5 [128, 135] to maximize the likelihood that the edges in the resulting network models represent genuine relations. Estimating a GGM with the *glasso* algorithm in combination with the EBIC model selection has been shown to reliably retrieve the true network structure [135] and is currently the dominant method for estimating GGMs in psychological network estimation [103, 104, 112].

The networks were visualized with the R package *qgraph* [132]. The node locations were determined using a modified version of the Fruchterman–Reingold algorithm [136, 137] for

weighted networks [132], to ensure that strongly connected nodes with many edges in common are placed close to one another.

**2.5.3 Network analysis.** Once a network has been computed, different methods can be used to analyze its structure. Visual inspection is a useful first step that provides relevant information with minimal effort, especially for small networks [138, 139]. A more formal analysis of the relative importance of nodes in a network can for instance be performed by quantifying their direct (strength centrality) or indirect (closeness centrality) connectivity with other nodes or their mediating capacity between other nodes (betweenness centrality). When two networks need to be compared, their layout should be constrained to allow visual comparison (e.g. by using the *averageLayout* option in the *qgraph* R package: [132]) and permutation tests can be used quantify their structural similarity (e.g., by using the R package NetworkComparisonTest: [140]). Each of these steps will be discussed in the next sections.

*2.5.3.1 Centrality indices.* The importance of an individual node in a network is reflected in the number and strength of its connections to other nodes. In network analysis this is generally operationalized through three centrality indices: node strength (quantifying how strongly a node is *directly connected* to other nodes), closeness (quantifying how strongly a node is *indirectly connected* to other nodes), and *betweenness* (the number of times a node lies on the shortest path between two other nodes [130, 141]). To investigate the extent to which the individual variables (nodes) in our models play a mediating role in odor-evoked affect, we used the *centrality_auto* function to compute their strength, closeness and betweenness indices and we visualized the results (as z-scores to ensure comparability between networks) with the *centralityPlot* function, both from the R package *qgraph* [132]. Node strength is computed as the sum of the absolute weights of all edges connected to a node. A strength-central node is one that strongly affects other nodes. The closeness centrality of a node indicates the average distance from all other nodes in the network and is computed as the inverse of the sum of the shortest distances between the node and all other nodes. A closeness-central node is affected strongly (either directly or indirectly) by other nodes in a network. The betweenness centrality of a node is computed as the number of times that the node is on the shortest path between any two other nodes. A betweenness-central node connects a large number of other nodes, serving a bridge function. We quantified the stability of the centrality indices by their correlation stability (CS) coefficient, the value of which should preferentially exceed 0.5 [112].

*2.5.3.2 Accuracy and stability.* We used the R package *bootnet* [112] to evaluate the robustness (in terms of accuracy and stability) of the estimated networks through a nonparametric bootstrap sampling procedure [142].

First, we assessed the accuracy of the edge weights by computing and plotting the 95% confidence intervals (CIs) for each edge from a distribution of edge weights generated by sampling the data 1,000 times with replacement [107, 112, 143].

Next, we evaluated the stability of the networks by repeatedly correlating the centrality indices of the original data with the centrality indices calculated from subsamples comprising progressively fewer cases. The number of bootstraps was again set to 1,000. A centrality index is considered less stable when its correlation value decreases with a reduction of the sample size. This is quantified by the correlation stability coefficient (CS-coefficient), which represents the maximum proportion of cases that can be dropped while maintaining 95% probability that the correlation between the centrality index of the full dataset and that of the subset is at least .70 (denoted as CS(cor = .70); the value of .7 was chosen since this is typically regarded as a large effect: [144]). CS-coefficients above .5 indicate a high stability, while a minimum CS-coefficient of .25 is recommended for sufficient stability to warrant further interpretation of the centrality indices [107, 112].

**2.5.4 Network comparison.** We compared the structure of the unpleasant (UOEA) and pleasant (POEA) odor evoked affect networks in several ways.

First, we performed a visual comparison between the UOEA and POEA networks. Then, we computed a similarity index by correlating the edge weights across the two networks (i.e., by correlating their regularized partial correlation matrices: [145]). This index measures the correspondence between the strength of the network connections in both models. If the correlation equals one, the connections in both networks are perfectly linearly related, meaning that both networks essentially have the same structure; if it equals zero, the networks have no detectable linear correspondence; if it equals minus one, the networks are exact opposites.

Next, we formally tested their difference using the R package *NCT* (Network Comparison Test: [146]). The NCT is a two-tailed permutation test in which the difference between two groups is calculated repeatedly (10,000 times) for randomly regrouped cases. This produces a distribution of values under the null hypothesis (i.e., assuming equality between the groups) enabling one to test whether the observed difference in global strength differs significantly ($p < .05$) between two networks. The NCT can test invariance of structure and invariance of global strength. Invariance of structure is tested by comparing the largest observed difference (M) between corresponding edges in the two networks to that observed under permutation. Invariance of global strength (S) is tested by comparing the value of this parameter to that observed under permutation. Previous network research has shown that strength is typically the most stable and reliable centrality index [128, 143].

## Results

Four participants were identified as outliers (their standardized ratings exceeded two standard deviations from the mean). Two of them gave an extremely low valence rating for the peach odor (ID = 31, see Table 1). Two other participants gave extremely low ratings for either the intensity of the pineapple odor (ID = 21) and the familiarity of the peppermint odor (ID = 2). After excluding these four participants from further analysis the remaining sample consisted of 52 participants (31 females and 21 males, with a mean age of 24.3 years, SD = 4.7).

First, we determined the mean ratings for valence, arousal, familiarity and intensity for each odor over all participants. The results are listed in Table 1. Fig 4 shows that the overall relation between mean valence and arousal can be described by a U-shaped (quadratic) form: odors scoring near neutral (zero) on mean valence have the lowest mean arousal ratings, while odors scoring either high (pleasant) or low (unpleasant) on mean valence show higher mean arousal ratings. Hence, odors with opposite mean valence ratings may yield similar mean arousal ratings. Because of the functional dichotomy that may exist in the relation between valence and the other variables that are measured in this study (e.g., the relation between valence and familiarity: [25]), we separately analyzed the results for unpleasant (odors with negative mean valence ratings) and pleasant (odors with positive mean valence ratings) stimuli. We classified the 16 odors with mean valence ratings below neutral as unpleasant stimuli (the odors with ID: 4, 7, 11, 12, 14, 15, 19, 21, 26, 27, 28, 35, 37, 38, 39, 40; see Table 1 and Fig 4) and the 24 odors with mean valence ratings above neutral as pleasant stimuli (the odors with ID: 1, 2, 3, 5, 6, 8, 9, 10, 13, 16, 17, 18, 20, 22, 23, 24, 25, 29, 30, 31, 32, 33, 34, 36). The mean intensity ratings listed in Table 1 show that the set of odor stimuli contained no outliers in intensity: all intensities were within about 28% of the mean intensity (which closely agrees with the criterion of 25% set by the developers of the Sniffin' Sticks [116]).

Next, we estimated two network models: one for pleasant odors and one for unpleasant odors. In the following we will use the previously introduced abbreviations for the variable names (see Fig 1) to designate each node in these networks: BV and BA indicate respectively

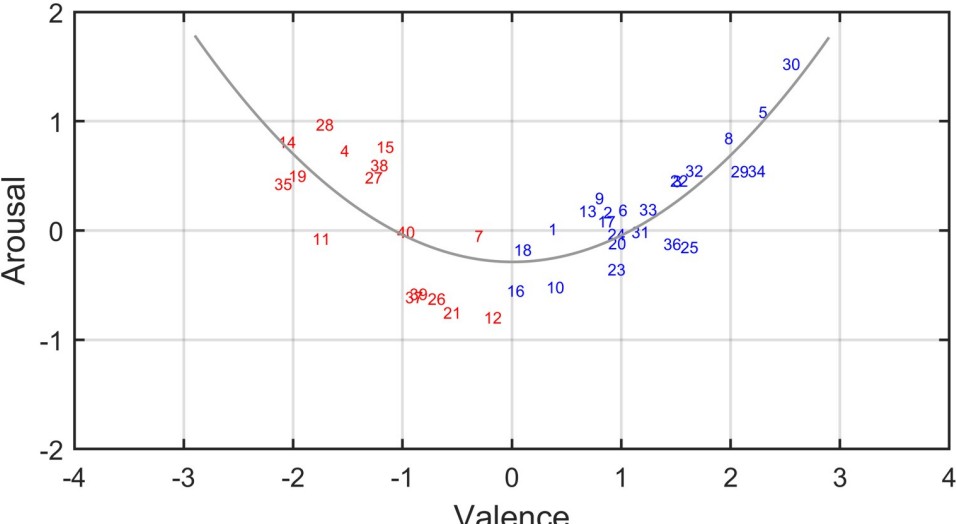

**Fig 4. Relation between mean valence and arousal ratings for the odors used in this study.** The numbers correspond to the identifiers in Table 1. Red numbers correspond to odors that received a negative mean valence rating, while blue numbers indicate odors that received a positive mean valence rating. The gray curve represents a quadratic fit to data points ($R^2$ = .59).

the valence and arousal components of the participant's baseline affective state (measured at the start of the experiment), T designates the detection threshold, F represents the familiarity of an odor, I its intensity, while V and A represent respectively the valence and arousal associated with an odor. Unpleasant and pleasant odor-evoked affect will be referred to as UOEA and POEA respectively.

## 3.1 Network estimation

Fig 5 shows a graphical representation of the estimated (regularized) partial correlation network models (Gaussian Graphical Models) for (Fig 5a) unpleasant odor evoked affect (UOEA) and (Fig 5b) pleasant odor evoked affect (POEA), based on the sample of 52 participants that evaluated 16 unpleasant and 24 pleasant odors. Table 2 lists the partial correlations between the different variables in both networks. The resulting network structures are parsimonious due to the LASSO estimation: the UOEA and POEA networks (each containing 7 nodes) respectively have only 6 (3 positive and 3 negative) and 7 (5 positive and 2 negative) non-zero edges out of the 21 (= 6*7/2) possible edges.

## 3.2 Network analysis

**3.2.1 Centrality indices.** Both resulting (UOEA and POEA) networks consist of three independent (unconnected) components: a trivial graph consisting of one node (BA), a simple graph consisting of two nodes (BV and T), and a connected graph consisting of the remaining four nodes (A, F, I and V). Table 3 lists the three (standardized, z-scored) centrality indices (strength, betweenness and closeness) for the nodes in the largest (4-node connected component of both the UOEA (Fig 5a) and POEA (Fig 5b) network models. It appears that these nodes differ substantially in their centrality estimates.

I is the most central node in the UOEA network, with the highest scores on all three centrality indices. This implies that I most significantly (directly and indirectly) contributes to the

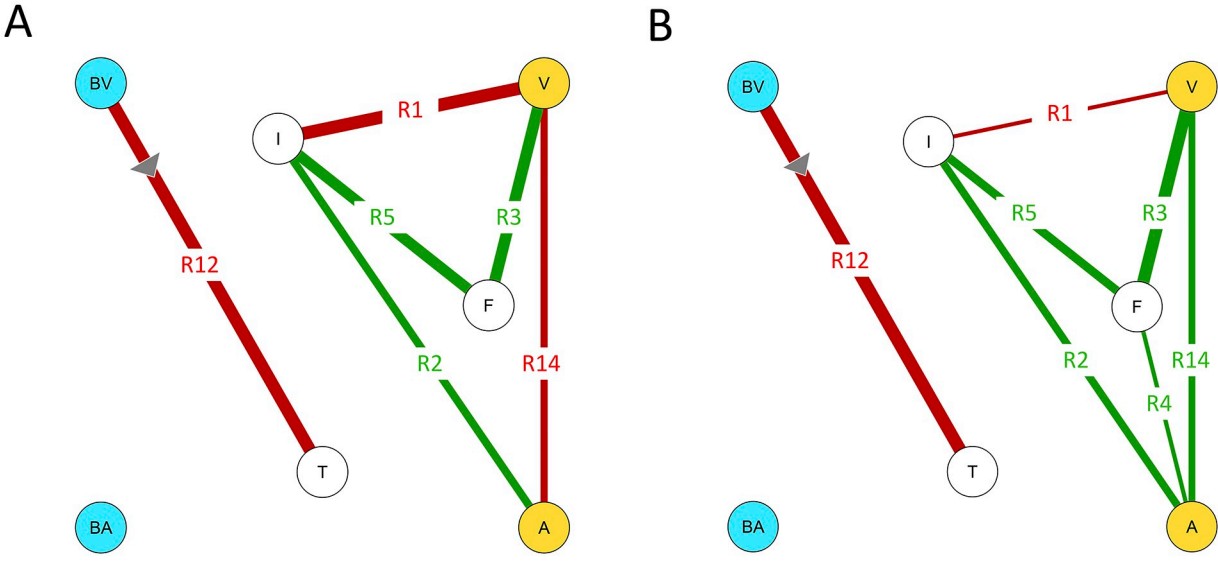

**Fig 5. Estimated partial correlation networks for unpleasant (a) and pleasant (b) odor evoked affect, based on a sample of 52 participants that evaluated 16 unpleasant and 24 pleasant odors.** Nodes represent the observed variables (for the meaning of the node labels we refer to the text), while green and red edges represent positive and negative partial correlations. Edge labels represent the relational identifiers from Fig 1. The width of the edges increases with the magnitude of the correlations and is scaled to the strongest edge (and therefore not comparable between graphs in an absolute sense).

affective appraisal of unpleasant odors. F scores highest on all three centrality indices in the POEA network, indicating that this factor significantly contributes to the affective appraisal of pleasant odors.

**3.2.2 Accuracy and stability.** The accuracy and stability of the centrality indices was investigated by a case-dropping bootstrapped sampling procedure with 1,000 samples.

Fig 6 shows the bootstrapped 95% confidence intervals for the edge-weights in the estimated UOEA and POEA networks. This figure shows that the confidence intervals of positive and negative edges do not overlap, meaning that edges with opposite signs in Fig 5 are

**Table 2. Partial correlations between the different variables in the estimated UOEA and POEA networks (Fig 5).** Dashes represent relations from the HOEA model that do not appear in the estimated networks.

| Label | Relation | UOEA | POEA |
|-------|----------|------|------|
| R1 | I – V | -0.36 | -0.11 |
| R2 | I – A | 0.25 | 0.24 |
| R3 | F – V | 0.32 | 0.40 |
| R4 | F – A | - | 0.12 |
| R5 | F – I | 0.31 | 0.27 |
| R6 | BV – V | - | - |
| R7 | BV – A | - | - |
| R8 | BA – V | - | - |
| R9 | BA – A | - | - |
| R10 | BV – I | - | - |
| R11 | BA – I | - | - |
| R12 | BV – T | -0.42 | -0.42 |
| R13 | T – I | - | - |
| R14 | V – A | -0.18 | 0.21 |

**Table 3. Strength, closeness and betweenness centrality indices (using standardized *z*-scores to facilitate interpretation) for each node in the estimated networks (see Fig 5) for unpleasant and pleasant odor evoked affect, estimated from 1,000 bootstrap replications with the adaptive LASSO algorithm. Maximal values are printed in boldface.**

| Node | Unpleasant odors | | | Pleasant odors | | |
|---|---|---|---|---|---|---|
| | Betweenness | Closeness | Strength | Betweenness | Closeness | Strength |
| A | -0.38 | -1.17 | -0.29 | -0.59 | -1.43 | 0.25 |
| F | -0.38 | -0.39 | 0.42 | **1.46** | **0.68** | **1.09** |
| I | **2.27** | **1.14** | **1.27** | -0.59 | 0.07 | 0.43 |
| V | -0.38 | 0.42 | 1.07 | **1.46** | **0.68** | 0.83 |

significantly different. However, the edge weights of some positive edges in Fig 5 (e.g., R2, R5, R14) may not be significantly different on the population level, since their confidence intervals show a large degree of overlap.

Fig 7 shows the stability plots for the centrality indices strength and betweenness (note that closeness could not be evaluated because of the infinite distance between the unconnected components) for both the unpleasant (Fig 7a) and pleasant (Fig 7b) odor evoked affect network models from Fig 5. These figures show that node strength (the associations of a node with its immediate neighbors) is highly stable for variations in sample size in both networks. Betweenness (connecting other nodes) shows a somewhat steeper decrease in accuracy with sample size than strength, especially in the POEA network. As a result, we cannot confidently conclude that any node in the POEA network is significantly more central than any other.

Table 4 lists the correlation stability coefficients for the network centrality indices. Strength centrality is the only stable network characteristic. For both networks (UPOEA and POEA), the strength stability coefficient is 0.75, exceeding the recommended minimum value of 0.25. This means that strength centrality induces a meaningful order on the nodes in the networks.

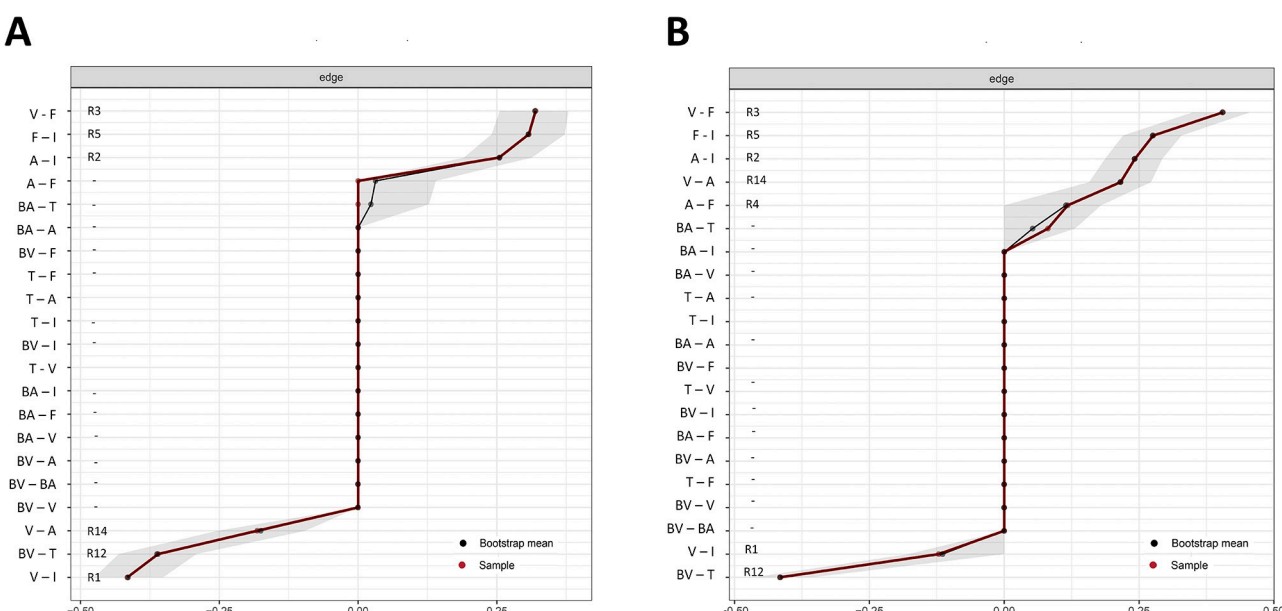

**Fig 6. Bootstrapped 95% confidence intervals gray areas for the edge-weights in the estimated networks for unpleasant (A) and pleasant (B) odor evoked affect.** The red line connects the sample values, the black line the bootstrap means. The gray area represents the CIs. Each point represents one edge in the network, ordered from the edge with the highest weight (top) to the edge with the lowest weight (bottom). The labels along the outside of the vertical axis indicate the relations between different variable pairs (see text for the abbreviations) while the corresponding labels on the inside correspond to the edge labels relation numbers in the hypothetical odor evoked affect HOEA model (see Fig 1).

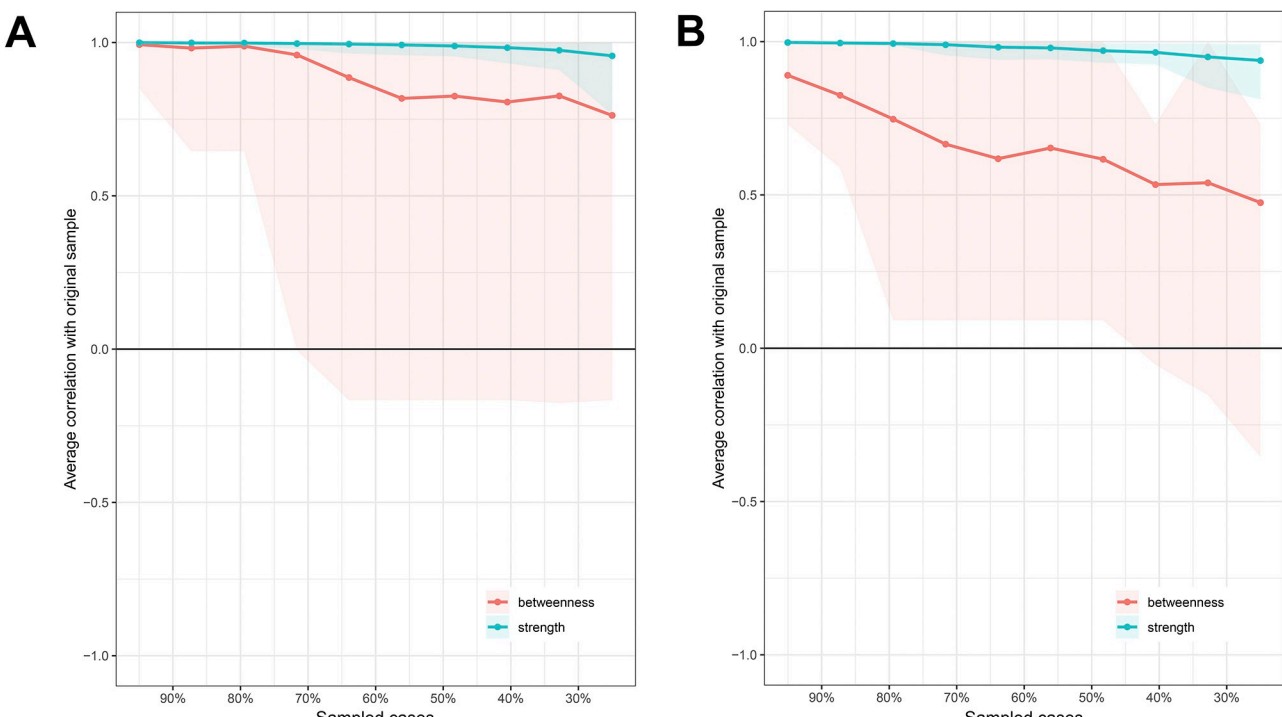

**Fig 7. Stability of the central indices strength and betweenness of the estimated networks for unpleasant (A) and pleasant (B) odor evoked affect.**
Data points represent the average correlation between the estimates based on subsamples, expressed as a percentage of original number of cases and the entire original sample. Areas indicate the range between the 2.5th and 97.5th quantiles.

## 3.3 Network comparison

**3.3.1 Global network structure.** A visual comparison of the UOEA and POEA networks in Fig 5 shows that their structure is very similar: six edges occur in both networks (R1, R2, R3, R5, R12, R14). This observation is confirmed by the Pearson correlation between the adjacency matrices of both networks, which is r = .81, indicating a high degree of similarity. Also, the NCT revealed that the global strength of the UOEA network (1.85) does not differ significantly from that of the POEA network (1.75; p = .56). Corresponding relations in both networks have similar absolute partial correlation strengths (see Table 2) and the same polarity, except for the relation R14: as expected (see section 1.3) valence and arousal are positively correlated for positive odors and negatively correlated for unpleasant odors.

The main difference between both network structures is the relation between F and A (R4): familiarity only appears to (positively) influence the arousing quality of positive odors, but not of negative odors.

**3.3.2 Intensity.** In both emerging network models, odor intensity is negatively correlated with valence in (R1): the more intense being the more unpleasant. However, the relation is stronger for unpleasant odors than for pleasant odors. Odor intensity is positively correlated

**Table 4. Correlation stability coefficients for the network centrality indices.** The CScor = 0.7 coefficients represent the maximum proportion of cases that can be dropped such that the correlation between the original centrality metric and those of the sampled subsets is 0.7 or higher with 95% probability.

| Network model | Strength | Closeness | Betweenness |
|---|---|---|---|
| UOEA | 0.75 | 0.00 | 0.13 |
| POEA | 0.75 | 0.00 | 0.13 |

with subjective arousal (R2): odors that are more intense are rated as more arousing, independent of odor valence. These results both agree with the general findings in the literature, as embodied in both HOEA models (Fig 1).

**3.3.3 Familiarity.** Familiarity and valence are strongly positively correlated (R3): the more familiar an odor, the more pleasant it is judged, independent of odor valence. Familiarity is weakly positively correlated with odor evoked arousal for pleasant odors (R4), while no relation emerges for unpleasant odors. This is in contrast with the HOEA model, that predicts a negative correlation between odor familiarity and odor evoked arousal, independent of odor valence. In agreement with the HOEA model, familiarity is positively correlated with odor intensity in both networks (R5): participants perceive familiar odors as more intense than unfamiliar odors, independent of odor valence.

**3.3.4 Affective state.** The current results show no modulating effect of the arousal and valence components of the observer's baseline affective state on both the intensity and the affective appraisal of odors, independent of their valence (relations R6-R11 from the HOEA model are absent in the UOEA and POEA models).

In agreement with the HOEA model, the valence component of baseline affective state (BV) correlates strongly and negatively with the odor detection threshold (R12). This agrees with the finding that negative mental states reduce odor sensitivity [54, 86], whereas a positive mental state can enhance odors sensitivity [86]. Contrary to our expectations, we find no negative correlation between T and I (R13).

## 4 Discussion

Based on a review of the literature we identified four individual factors that can influence odor-evoked affect as measured in term of valence and arousal: baseline affective state, odor sensitivity (detection threshold), odor intensity and odor familiarity. However, the exact nature of the relations between each of these variables and their influence on odor evoked affect are still largely unknown. To investigate these relations, we first constructed a hypothetical relational model, based on the small amount of literature that is currently available. Then we performed an observer experiment to collect data that can be used to verify this model. We used network analysis to explore the relations between the measured variables and odor-evoked affect through (regularized) partial correlations. This technique offers a data-driven view of the salient relationships between the variables of interest: relations emerge as partial correlations between the individual variables. Since the resulting GGMs are undirected networks it is not possible to discern causal relations. However, the absence of an edge between two factors in these models provides strong evidence that neither factor causes the other.

Because of the functional dichotomy that may exist in the relation between valence and the other variables that are measured in this study (e.g., the relation between valence and familiarity: [25]), we investigated the results for unpleasant (odors with negative mean valence ratings) and pleasant odors (odors with positive mean valence ratings) separately. Hence, we obtained two models: one for unpleasant odor evoked affect (UOEA model) and one for pleasant odor evoked affect (POEA model). It appears that both models are highly similar. The positive correlations between F and V for pleasant odors [12, 51–53, 67–73], and between F and I [52, 70], and between I and A [12, 14, 56, 57] that have been reported in the literature, also emerge in both networks. Other known relations, such as the negative correlations between BV and T [86, 147, 148], and between I and V [55], also consistently emerge in both models. The main difference between both network structures is the relation between F and A (R4): familiarity only appears to contribute to the arousing quality of positive odors, but not of negative odors. The similarity between the UOEA and POEA networks suggests the existence of multiple

affective modes [149]. This notion is reflected in bivariate models of valence (e.g., [150]) that represent pleasant and unpleasant feelings along two separate unipolar dimensions [151–153]. Future studies on mixed affective responses to odors may provide more evidence about the dual nature and characteristics of the systems mediating the affective appraisal of odors [154].

We found that the valence component of baseline affective state correlates strongly and negatively with the odor detection threshold (R12). Although this result agrees with the findings that (1) negative mental states raise odor detection thresholds [54, 86] and (2) a positive mental state can lower detection thresholds [86], this relation has (to the best of our knowledge) not been reported previously. Future studies are needed to investigate whether this relation can be replicated with different odors and populations.

In contrast to previous studies that reported no consistent relation between familiarity and valence (R3) for unpleasant odors [25, 33, 51], we find that R3 is positive, independent of odor valence. This discrepancy most likely arises from the fact that these earlier studies computed the correlations at the group level, while we use individual standardizing (i.e., the data for each individual is standardized before computing the networks, to retain a within-individual approach to the data). Thus, it appears that at the individual level, increasing familiarity (i.e., a reduction of uncertainty) consistently enhances valence, both for pleasant odors (valence becomes more positive) and for unpleasant ones (valence becomes less negative). This agrees with the general tendency to attribute more weight to affective information in conditions of uncertainty [155]. Unpleasant odors that are unfamiliar (i.e., for which it has not yet been established whether they are harmful) may be rated as more unpleasant than unpleasant odors that are more familiar and known to be harmless. Our current finding also seems to agree with the finding that the unpleasant odors of fish and garlic were rated as less unpleasant by children who correctly identified them [122].

Contrary to our expectations, we found no negative correlation between T and I (R13).

The high degree of centrality of I in the UOEA network model suggests that I is the most crucial factor influencing the affective appraisal of unpleasant odors. The same holds for F in the POEA network model. T is only affected by BV and appears to have no effect on any of the other variables.

Partial correlation networks are exploratory hypothesis-generating structures that are merely indicative of potential causal effects [104]. With this caveat in mind, the emerging association between BV and T can still be interpreted as a causal relation, due to the temporality of the associated measurements (i.e., the assessment of BV precedes the measurement of T). Insofar as the centrality of a node can be taken to reflect the causal connections emanating from that node, it appears that I dominates the affective appraisal of unpleasant odors (R1, R2), while familiarity dominates the affective appraisal of pleasant odors (R3, R4).

## 4.1 Limitations

Several limitations should be acknowledged for the present study.

Except for the edges connecting the baseline affective state (BV and BA), the edges in our networks of partial correlation coefficients are undirected and therefore preclude any conclusions about causal (unidirectional or reciprocal) relations. Although correlation does not establish causation, it is consistent with it. Also, the absence of an edge between two variables provides evidence that they are not causally related. Hence, although our exploratory models are in no way confirmatory of causal relationships, they can serve to inspire targeted experimental studies investigating possible predictive relationships. Future research could expand network analysis with a Bayesian network approach [143, 156] to investigate the causal relationships between the different parameters involved in affective odor perception.

In this study we measured olfactory sensitivity using the validated Sniffin' Sticks based threshold test (SST), which has become a popular (validated and standard) procedure in the literature. This test was also used in several references cited in this study, allowing us to relate our current finding to those earlier results. However, the SST is a single-molecule based general diagnostic tool to assess olfactory functioning. Even for people with a normal sense of smell, sensitivity can vary significantly between individual odors [157]. Hence, the SST measurements cannot be translated to different odors [157] and the test cannot distinguish between general smell dysfunction and PEA insensitivity [158]. This may be the reason why no relation between T and I appears in our results. Future studies could investigate the relationship between threshold and intensity more closely by measuring a detection threshold for each individual odor that is used. However, such a procedure will be very time consuming and tiresome for the participants. Alternatively, future studies may also consider to obtain olfactory threshold assessments that are less dependent on the individual variability in sensitivity to specific odorants by using a threshold test based on complex odor mixtures (e.g. SMELL-R: [158]; see also [159]). However, this invariance only appears to hold for specific odor mixtures [160], and there is currently no generally accepted and validated complex-odor based threshold test available.

The absence of connections between observed variables (nodes) in our networks can either imply that these variables are statically independent when conditioning on all other variables, or it can mean that there was simply insufficient power to detect a relation between these variables [161]. The betweenness centrality estimates were insufficiently stable and should be therefore interpreted cautiously (i.e., the order induced on the nodes by betweenness is not very meaningful). The stability of betweenness centrality might have been greater if we had tested more participants. Future studies including a larger number of participants are required to resolve these issues.

Using Gaussian graphical models, we implicitly assumed that the variables in our models are linearly related. Diagnostic scatterplots show that this assumption is met within the groups of pleasant and unpleasant odors. Although the use of a different kind of correlation estimate (e.g., Spearman of Kendall) would allow for modelling non-linear relations (such as the one shown in Fig 4) these models would be less optimal and new methods are needed to construct a fitting network model.

The current study is a first attempt to construct a network representation for some the main factors that significantly influence the affective appraisal of odors. Given that we mainly focused on a limited set of mediators (intensity, familiarity, sensitivity and the baseline affective state of the observer), we may have missed other relevant factors. For instance, we did not investigate the effects of other factors known to influence the affective appraisal of odors, such as attention [19] or inter-individual variations like differences in physiological state (hunger, satiety: [162]), sex [163], age, semantic knowledge and cultural background [73, 99]. Future studies should investigate how these factors affect network models of odor evoked affect.

In conclusion, these limitations notwithstanding, this study demonstrates that psychometric network analysis can be an effective technique towards the construction of an integral model for the relations between the various factors that influence the affective appraisal of odors. Such a model may constitute the basis for implementing targeted investigations of the way in a wide range of user characteristics determine the affective appraisal of odors.

## Supporting information

**S1 Data.**
(XLSX)

## Acknowledgments

The authors thank Jos van den Enden (Retroscent, https://retroscent.com) for kindly providing us with a set of additional odorant samples.

They are also grateful for the many comments and suggestions of the anonymous reviewers, that significantly helped to improve the quality of this study.

## Author Contributions

**Conceptualization:** Yingxuan Liu, Alexander Toet.

**Data curation:** Yingxuan Liu, Alexander Toet, Sophia Eijsman.

**Formal analysis:** Yingxuan Liu, Alexander Toet, Tanja Krone, Robin van Stokkum, Jan B. F. van Erp.

**Investigation:** Yingxuan Liu, Alexander Toet, Sophia Eijsman.

**Methodology:** Yingxuan Liu, Alexander Toet, Tanja Krone, Robin van Stokkum, Sophia Eijsman, Jan B. F. van Erp.

**Project administration:** Yingxuan Liu, Sophia Eijsman.

**Resources:** Alexander Toet.

**Software:** Yingxuan Liu, Alexander Toet, Tanja Krone, Robin van Stokkum.

**Supervision:** Alexander Toet, Jan B. F. van Erp.

**Validation:** Alexander Toet, Tanja Krone.

**Visualization:** Yingxuan Liu, Alexander Toet.

**Writing – original draft:** Yingxuan Liu, Alexander Toet, Sophia Eijsman, Jan B. F. van Erp.

**Writing – review & editing:** Yingxuan Liu, Alexander Toet, Tanja Krone, Robin van Stokkum, Sophia Eijsman, Jan B. F. van Erp.

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
