## [Decision Letter · Decision Letter 0]

20 Mar 2020

PONE-D-19-24559

A network model of affective odor perception

PLOS ONE

Dear Dr. Toet,

Thank you for submitting your manuscript to PLOS ONE. After careful consideration, we feel that it has merit but does not fully meet PLOS ONE’s publication criteria as it currently stands. Therefore, we invite you to submit a revised version of the manuscript that addresses the points raised during the review process.

The study and the approach presented by the authors is interesting, however, the reviewers highlighted the need for deep revision of the introduction and the methodology (that can affect their results) and raised doubts on specific aspects of the experimental protocol. The authors should follow the reviewers' suggestions and revised the paper accordingly.

We would appreciate receiving your revised manuscript by the 20th of June. To enhance the reproducibility of your results, we recommend that if applicable you deposit your laboratory protocols in protocols.io, where a protocol can be assigned its own identifier (DOI) such that it can be cited independently in the future. For instructions see: http://journals.plos.org/plosone/s/submission-guidelines#loc-laboratory-protocols

We look forward to receiving your revised manuscript.

Kind regards,

Alberto Greco

Academic Editor

PLOS ONE

Journal Requirements:

2. We note that Figure 2 in your submission contain copyrighted images. All PLOS content is published under the Creative Commons Attribution License (CC BY 4.0), which means that the manuscript, images, and Supporting Information files will be freely available online, and any third party is permitted to access, download, copy, distribute, and use these materials in any way, even commercially, with proper attribution. For more information, see our copyright guidelines: http://journals.plos.org/plosone/s/licenses-and-copyright.

1.    You may seek permission from the original copyright holder of Figure 2 to publish the content specifically under the CC BY 4.0 license.

Reviewers' comments:

Reviewer's Responses to Questions

**Comments to the Author**

1. Is the manuscript technically sound, and do the data support the conclusions?

Reviewer #2: Partly

Reviewer #3: Yes

2. Has the statistical analysis been performed appropriately and rigorously? 

Reviewer #2: No

Reviewer #3: I Don't Know

3. Have the authors made all data underlying the findings in their manuscript fully available?

Reviewer #2: Yes

Reviewer #3: Yes

4. Is the manuscript presented in an intelligible fashion and written in standard English?

Reviewer #2: Yes

Reviewer #3: Yes

5. Review Comments to the Author

Reviewer #2: The aim of the authors is to systematically investigate the relationships between affective and perceptive judgements of odors, mood and olfactory capacities. Unfortunately and for the reasons I develop below, I do not think that they can claim a substantial contribution for this purpose.

First of all, the tool they use to measure mood is not suitable for fine mood measurement. The authors should better justify the use of EmojiGrid to characterize the mood of participants. It is obvious that positioning one's mood on a two-dimensional space of valence and arousal allows to obtain global information on this mood. But this same space reduces the possibility to dissociate qualitatively different mood states. This is why standardized mood questionnaires such as POMS (Profile of Mood States) exist, which contains more than two valence and arousal scales. Using the EmojiGrid, how to dissociate a state of anxiety (negative valence and high arousal) from an irritable state (negative valence and high arousal). I think you would agree that these moods are not identical and could influence olfactory perception in a specific way. This is problematic because the authors aim to make the link between mood, olfactory perception and olfactory emotional response. How can they answer this question if they do not measure mood according to scales that capture best the quantitative and qualitative differences in mood?

See for instance: “Profile of Mood States.” McNair, D. M., Lorr, M., & Droppleman, L. F. (1992). Revised Manual for the Profile of Mood States. San Diego, CA: Educational and Industrial Testing Services., no. 65 (37),

Second, using the detection threshold of a single molecule in the model is problematic. The authors determine the AEP detection threshold for each participant and use this value to establish the influence of the threshold on odour-related emotional response. This could have been a very important contribution if the authors had performed the detection threshold for each odour used in the affective assessment (I can absolutely not blame the authors for not doing it because it would be an extremely tedious job). Thresholds are specific to each molecule; determining a perception threshold for one molecule does not predict thresholds for others. The test used for the threshold is effective in clinical investigation to detect possible anosmia or hyposmia. This allows the patient to be quickly located according to a known standard. But the approach of the authors in this research is to observe systematic relations between detection thresholds, perceptive and affective dimensions thanks to the use of 40 different odorants. It is not by introducing the threshold to one and only one molecule that these systematic relationships can be studied. One argument for my criticism is that the network analysis the authors performed does not reveal a relationship between threshold and perceived intensity. If the threshold measurement with one molecule was a good estimate of overall olfactory capacity, then a strong positive relationship would have been expected.

Hsieh, J. W., Keller, A., Wong, M., Jiang, R. S., & Vosshall, L. B. (2017). SMELL-S and SMELL-R: olfactory tests not influenced by odor-specific insensitivity or prior olfactory experience. Proceedings of the National Academy of Sciences, 114(43), 11275-11284.

There is, in my opinion, a set of theoretical or definitional clarifications that should be made in the introduction to make the authors' purpose clearer to the reader:

Line 27: Just because there is a high degree of overlap between the cerebral systems that process olfaction and emotions does not mean that the responses have to be "various", this overlap could exist with only one emotional response. Please rephrase or delete that attempted explanation.

Line 30: These dimensions are not necessarily the most fundamental, they are fundamental but not the only ones. They are especially the most used to measure the "core affect". Fontaine, J. R., Scherer, K. R., Roesch, E. B., & Ellsworth, P. C. (2007). The world of emotions is not two-dimensional. Psychological science, 18(12), 1050-1057.

Line 32 and following: I don't understand why the authors use the term "determined" in this sentence, it suggests that there is the valence and arousal of smell (are these intrinsic properties of smell?) that cause the emotional response. Valence and arousal are dimensions of the affective response. The authors may mean that it is possible to describe or characterize the affective response through the psychological dimensions of valence and arousal. There is abundant literature that describes the emotional response as multi-componential, with its subjective, physiological, motivational, cognitive and expressive dimensions. The emotional response cannot be limited to the subjective dimensions of valence and arousal. I suggest that the authors reduce their rationale to the characterization of the relationships between the subjective dimensions of olfactory evaluations.

It seems that the authors differentiate between the intensity of odour, which would be a subjective quality and which they name “perceived intensity”, while they might suggest that the other dimensions represent intrinsic properties of odour, which of course are not. I think it is important to clarify that from the moment that participants are asked to evaluate odour on any dimension, this dimension is subjective and deserves the qualifier "perceived". Thus, the authors could remove the mention of “perceived” in “perceived intensity” or, on the contrary, add this mention to the other dimensions.

Concerning the data availability and analysis strategy.

1. All edges should be undirected except those from Mood which should go from mood to the other variables. Authors can’t postulate mutual relationship in this case because they don’t measure the effect of each odor on participant’s mood.

2. Authors used Gaussian graphical model but didn’t give evidence there data continuous data are distributed. By deliberately selecting a group of positive and a group of negative odours, I fear that the authors have favoured non-normal valence distributions.

3. It seems to me that the measurements are repeated in the same individual (all smells are judged by each individual). Why didn't the authors take this factor into account when applying the GGM? In the model presented by the authors, it seems to me that each observation is considered unique and independent (cross-sectional data analysis) which is not the case.

4. I particularly appreciate the willingness of the authors to share the data from this experiment along with the analysis scripts. Indeed, this is a practice that must now be encouraged if science is to evolve in complete transparency. However, in the data shared by the authors, participant numbers and odours’ names are not reported, which limits the verification and exploitation of the data. I strongly encourage the authors to provide this information which will be very important for the olfactory research community.

Discussion.

The authors conducted the analyses separately for positive and negative odours to “avoid neutralization effects, and because of the functional dichotomy that may exist in the relation between valence and the other variables that are measured in this study (e.g., the relation between valence and familiarity: [37])” The results they present suggest that pleasant and unpleasant networks do not differ. What are the functional consequences of this result? What does this result tell us about the nature of the relationships between the subjective odour dimensions or the treatment of pleasant and unpleasant odours? There are still some small differences between the two networks but it is very difficult for the reader to imagine the importance of these differences.

Reviewer #3: The authors modelled the affective appraisal of odors using sensory and affective parameters using network analysis. This is an interesting approach, especially because current mood is taken into account with references to Clore and Schwarz. Although I feel a model of affective odor appraisal should probably include additional contextual factors going beyond mood; mood can be considered context, and so this is an important step towards an integral model on affective odor appraisal. The authors do point out that future models should include gender and personality, for example. The authors should acknowledge that this may change the relations in the model quite a bit. For example, risk perception from odor exposure or source of exposure will affect the relations.

I have a number of minor to major comments and concerns, that I list below:

Introduction

I am missing a bit of nuance in certain paragraphs like 1.2.1 on perceived intensity. For example, the suggested relation between detection thresholds, intensity and valence should receive more attention. One could interpret the paragraph as implying that individuals with high thresholds show lower intensity perception or different valence perception. In view of the importance of perceived intensity in this paper, please extend this paragraph to include a section on threshold, determinants of perceived intensity and their relation.

There is more attention going to familiarity in 1.2.2, but what I miss is a clear definition of familiarity and how it relates to recognition/identification because it is my impression these terms are mixed up later on in the paper. I would recommend the authors refer to Kösters misfit theory.

1.3 Valence and arousal

What is not addressed here (unless I missed something) is the importance of learning. The general notion in the (human) olfaction literature is that odor valence, with perhaps very few exceptions, is the result of learning, rather than “innate” or “labelled line”. So the context in which an odor was experienced will determine how that odor is liked. While for many individuals context may be similar (e.g. delicious foods) resulting in same direction of valence, the resulting valence may vary (considerably) between individuals. It is important to address this in the paper, to set the stage for the modelling. This is to complete the set of factors we would ideally deal with.

Methods

I am not an expert on network analysis. Since the authors themselves reflect on sample size and (lack of) statistical power in the Discussion, can they reflect on whether the sample size of 56 for this type of analysis? How does this reflect on model fit and related parameters? Have they done a power analysis? I realize testing participants on odor perception in the lab requires substantial effort.

Iso-intensity

Testing for iso-intensity of the stimulus set requires more than three authors sniffing the stimuli to agree on intensity. Typically, this would require preparing multiple concentrations of the stimuli, having panelists smell and rate them and titrate the concentrations to get to iso-intensity over multiple sessions. Can the authors elaborate on what they did to achieve iso-intensity?

However, I am confused as to the need for iso-intensity. Intensity is a factor in the model and therefore it needs to vary. Judging from the outcome, it does, which would contradict the earlier statement of iso-intensity. Why would you need equal intensity in the first place? Perhaps the authors can clarify this.

Table 1: Please include concentrations (% compound in diluent or otherwise) in the Table for replication purposes. What did the safety assessment of compounds like tar, turpentine, and diesel fumes entail?

Procedure (page 14):

- It is always advised to have repetitions on all odor ratings. However, unless I missed this, I guess with 40 odors this was not possible.

Results

- Please provide a Table with means, SD or thresholds and other ratings for all odorants.

- How are individuals who do not judge peach as pleasant, perceive pineapple as lowly intense and peppermint as not familiar, outliers? There is no wrong or right answer as to how pleasant peach should be. See my previous comment about learning. These responses are not “erroneous” (page 20) so please remove. Outlier exclusion is only acceptable if a standard from exclusion is given based on e.g. 2-3 SD’s, so if there is a statistical reason as you model would be radically different perhaps if you included those.

- I am surprised by the finding that mood valence correlates strongly and negatively with threshold. I do not recall having ever seen this relation between lower threshold and positive mood before, even when the authors cite reference 60. I feel it would be important to test the model on another data set of different individuals/odors, so on new data. Will these relations hold? It would have been stronger if separate data sets had been used for model development and testing. But perhaps this is not a common approach for this type of data-driven analysis? Can the authors address this in the Discussion?

Finally, it is “Utrecht University” and not “University of Utrecht” (www.uu.nl – English)

6. PLOS authors have the option to publish the peer review history of their article (what does this mean?). If published, this will include your full peer review and any attached files.

Reviewer #2: No

Reviewer #3: No

---

## [Author Response · Author response to Decision Letter 0]

13 May 2020

See our "Reply to Reviewers" with a pointwise reply to each comment

---

## [Decision Letter · Decision Letter 1]

15 Jun 2020

PONE-D-19-24559R1

A network model of affective odor perception

PLOS ONE

Dear Dr. Alexander Toet

Thank you for submitting your manuscript to PLOS ONE. After careful consideration, we feel that it has merit but does not fully meet PLOS ONE’s publication criteria as it currently stands. Therefore, we invite you to submit a revised version of the manuscript that addresses the points raised during the review process.

Despite the improvement of the revised paper, there are still some major critical issues to be addressed. Specifically, the authors should better justify their model assumptions and how they can compare their model with the hypothetic model based on the literature. Finally, they have to solve the other minor points raised by the two reviewers.

Please submit your revised manuscript by 30 July 2020 If you will need more time than this to complete your revisions, please reply to this message or contact the journal office at plosone@plos.org. Please include the following items when submitting your revised manuscript:

We look forward to receiving your revised manuscript.

Kind regards,

Alberto Greco

Academic Editor

PLOS ONE

Journal Requirements:

Additional Editor Comments (if provided):

Reviewers' comments:

Reviewer's Responses to Questions

**Comments to the Author**

1. If the authors have adequately addressed your comments raised in a previous round of review and you feel that this manuscript is now acceptable for publication, you may indicate that here to bypass the “Comments to the Author” section, enter your conflict of interest statement in the “Confidential to Editor” section, and submit your "Accept" recommendation.

Reviewer #2: (No Response)

Reviewer #3: (No Response)

2. Is the manuscript technically sound, and do the data support the conclusions?

Reviewer #2: Yes

Reviewer #3: Partly

3. Has the statistical analysis been performed appropriately and rigorously? 

Reviewer #2: Yes

Reviewer #3: I Don't Know

4. Have the authors made all data underlying the findings in their manuscript fully available?

Reviewer #2: Yes

Reviewer #3: Yes

5. Is the manuscript presented in an intelligible fashion and written in standard English?

Reviewer #2: Yes

Reviewer #3: Yes

6. Review Comments to the Author

Reviewer #2: I think the authors have done an important work to improve the quality of the manuscript. The results presented here make it possible to introduce this family of network analyses to a community of researchers that has not used these techniques much so far. Anyway, I still have a few points I would like to see discussed/modified by the authors before I recommend it for publication.

General major points:

Since the hypothetic model is based on the literature and the model obtained in this study on experimental data related to a sampling of specific odours and participants, how can a meaningful comparison be made? Are the differences not explicable only in terms of sampling? Moreover, the hypothetical model proposed by the authors does not separate pleasant from unpleasant odours (it is crucial for some relations like R3). This makes any comparison very difficult. Why not just removing the theoretical model just report the existing relationships. Or the theoretical model must be separated according to the valence of the odours.

The analyses performed are based on linear correlations between the variables. Don't the authors think that this forced them to exclude any relation of another order (quadratic etc..) and therefore: i) to have to separate pleasant from unpleasant odours and ii) to conclude that there are two different treatments for the different valences. In other words, does the conclusion that pleasant and unpleasant odours must have specific treatments not simply follow from the fact that the analytical techniques employed examine linear relationships? I think the authors should discuss this point (linearity)a little bit more within the limitations.

Specific points:

Line 30 to 36: The three sentences introduce the same notion. Consider rewording them to make the information less redundant.

Line 43 and following: I don't quite understand the authors' inference that "These physiological parameters are linked to arousal..." It is a fairly robust finding in the psychophysiological literature that electrodermal activity is sensitive to arousal but that cardiac variations can be related to valence. An inverted U-curve between the skin conductance responses amplitude and the valence is very often observed for smells but also for images and sounds. This is interpreted as a reflection that subjective arousal and subjective valence are not independent.

Alaoui-Ismaili, O., Robin, O., Rada, H., Dittmar, A., & Vernet-Maury, E. (1997). Basic emotions evoked by odorants: Comparison between autonomic responses and self-evaluation. Physiology and Behavior, 62, 713–720.

Alaoui-Ismaili, O., Vernet-Maury, E., Dittmar, A., Delhomme, G., & Chanel, J. (1997). Odor hedonics: Connection with emotional response estimated by autonomic parameters. Chemical Senses, 22, 237–248.

Bensafi, M., Rouby, C., Farget, V., Bertrand, B., Vigouroux, M., & Holley, A. (2002a). Autonomic nervous system responses to odors: The role of pleasantness and arousal. Chemical Senses, 27, 703–709.

Bensafi, M., Rouby, C., Farget, V., Bertrand, B., Vigouroux, M., & Holley, A. (2002b). Influence of affective and cognitive judgments on autonomic parameters during inhalation of pleasant and unpleasant odors in humans. Neuroscience Letters, 319, 162–166.

Delplanque, S., Grandjean, D., Chrea, C.,Coppin, G.,Aymard, L.,Cayeux, I.,etal. (2009).Sequential unfolding of novelty and pleasantness appraisals of odors: Evidence from facial electromyography and autonomic reactions. Emotion 9, 316–328.doi:10.1037/a0015369

Lang, P. J., Greenwald, M. K., Bradley, M. M., & Hamm, A. O. (1993). Looking at pictures: Affective, facial, visceral, and behavioral reactions. Psychophysiology, 30, 261–273.

Pichon, A., et al. Sensitivity of Physiological Emotional Measures to Odors Depends on the Product and the Pleasantness Ranges Used. In: Frontiers in Psychology, 2015, vol. 6. doi: 10.3389/fpsyg.2015.01821

Line 56 and following: I confess to having some difficulty with the use of the term "influence" in sentence propositions such as "Factors influencing the affective appraisal of odors". This type of sentence implies a causal relationship between factors that would be primary and the affective appraisal of odour that would depend on these factors. I don't think it is possible to postulate this kind of causal relationship, and the experiment conducted by the authors and the partial correlation analyses do not allow us to infer a causal link. At best, the authors can examine the variables that covariate or correlate with the affective appraisal. Is an odour judged more intense then another because it is appraised as more unpleasant? Or is it the contrary? I will come back to this hidden “causal” concept later in my review.

Line 102: The sentence does not give a clear definition of familiarity. I also have difficulty understanding what the authors mean when they write: “Familiarity is implicitly linked to the affective and hedonic appreciation”. Initially, the authors seem to differentiate between affective and hedonic, but there is no mention of differences in the manuscript. Is it necessary to make this distinction? Second, can familiarity be defined without resorting to the hedonic value? Here is a definition that the authors can put in the context of olfaction: “A form of remembering in which a situation, event, place, person, or the like provokes a subjective feeling of recognition and is therefore believed to be in memory, although it is not specifically recalled.”

Line 114: “This suggest that pleasant and unpleasant odors are evaluated by different processing channels, in agreement with models that suggest that different evaluative channels are involved in the processing of negatively and positively valenced stimuli” This sentence doesn't add much with the two parts together. Please consider removing the first part.

Line 279: is it an arithmetic or geometric mean?

Line 388: is the parameter λ the sparsity parameter? Is so please mention it.

Line 419: Please check/correct the definition of betweenness.

Line 536, Table 2: Looking more closely (thanks to the excel file the authors provided) at the correlation between valence and familiarity for unpleasant odours only, conventional correlational analyses give no statistically significant relationship but the correlation coefficient is negative. This result is consistent with several studies in the literature cited by the authors failing to demonstrate any relationship between familiarity and valence for unpleasant odors. How then can we explain that a significant positive relationship appears in network analyses (R3, UPOEA) for the very same odors?

Line 548: typo () -> )

Line 557: For pleasant smells, familiarity and valence differ only in the strength parameter. How do we know if this is different enough to conclude that familiarity governs positive odour ratings? Is the answer at line 612 (“we cannot confidently conclude that any node in the PUOEA network is significantly more central than any other”)?

Line 597 and following: I have some difficulty understanding the authors' reasoning. Do they mean that the sample mean being included in the CI of the bootstrap mean, then the accuracy is good? And then because positive and negative edge so not share CI, they are clearly different? Please reformulate if possible.

Line 691: please add a nuance like “as measured in term of valence and arousal” because there are other emotion theories (not only dimensional models) and you could have chosen to assess affective appraisal via rating on basic emotions.

Line 718: The authors should not link the networks they highlight at the level of the subjective variables measured to any brain functioning. The structure of the networks they obtain can in no way reflect any brain functioning. Their conclusions should be limited to associative networks only.

Line 744 and following: I thought partial correlations can be indicative of potential causal pathways. As mentioned by Epskamp and Fried, 2018): “…partial correlation networks are thought of as highly exploratory hypothesis-generating structures, indicative of potential causal effects.” I think that the authors have to take precautions with the notion of causality. Even if it is very clear to them, they should be very careful not to imply that causal links are demonstrated by these analytical techniques. It is only suggested, assuming that there is no hidden latent unmeasured variable. Although the authors discuss this limitation later in the manuscript (in the limitation section). I think that they should not let people think that the causal link is there.

Epskamp, S., & Fried, E. I. (2018). A tutorial on regularized partial correlation networks. Psychological methods, 23(4), 617.

Reviewer #3: Intensities

The authors replied that the purpose of the pilot experiment was not to achieve iso-intensity.

I would agree that this would have been problematic in light of the objective of modelling. For the current objective, it is ideal for the odorants to cover a range of intensities so that there is variation. The authors state that extreme differences in intensity are undesirable and seem to aim for intensities to comparable (page 10). This leaves me in the dark as to the range of intensities they would deem acceptable or comparable. It now appears as if they “just went” with whatever the concentration was that the supplier sent them. Usually, dilutions would be made from those (in general high-intense) odorants the supplier will have prepared, to finetune the range of concentrations to cover a pre-agreed range of intensities.

I understand that with Sniffin’ Sticks already prepared by Burghart (with the odorant already in the Stick) it defies the purpose to have to go back to ordering the chemical and prepare the stick all over again, although these were prepared to be comparable and not vary too much. For odorants ordered from suppliers dilutions to achieve variation could have been made.

The authors have addressed the variation for valence, which clearly differed. Can the authors now address address the variation in intensity and if not by providing data from their pilot, by reflecting on this issue?

For an example of how concentrations determination is done to match intensities please see Keller, A., Hempstead, M., Gomez, I.A. et al. An olfactory demography of a diverse metropolitan population. BMC Neurosci 13, 122 (2012). https://doi.org/10.1186/1471-2202-13-122

for determination of “high” and “low” levels.

I would agree would be exaggerated to put this much work into determination of range for the present paper, but compared with this example I think the authors can do more than they currently do to address this. The means vary between 43 and 77 which strikes me as a bit narrow, but you can convince me otherwise.

Why not insert the table of means in the main paper?

Safety

When it comes to safety levels of tar and diesel fume, the authors now refer the reader to find this out for themselves on the supplier website. It is not the responsibility of the reader to verify safety, but that of the scientists. Based on the supplier information can the authors include a declaration in the paper that levels presented were safe? This is not to be dogmatic, but also to state that including such a statement to show that due diligence has been performed in protecting the participant’s health.

Finally:

I searched through the manuscript to find the difference baseline affective state and current affective state, but now conclude they are the same. If that is indeed the case, it would help to consistently use the same term.

7. PLOS authors have the option to publish the peer review history of their article (what does this mean?). If published, this will include your full peer review and any attached files.

Reviewer #2: No

Reviewer #3: No

---

## [Author Response · Author response to Decision Letter 1]

25 Jun 2020

See our "Reply to Reviewers" with a pointwise reply to each comment

---

## [Decision Letter · Decision Letter 2]

9 Jul 2020

A network model of affective odor perception

PONE-D-19-24559R2

Dear Dr. Alexander Toet,

We’re pleased to inform you that your manuscript has been judged scientifically suitable for publication and will be formally accepted for publication once it meets all outstanding technical requirements.

Kind regards,

Alberto Greco

Academic Editor

PLOS ONE

Additional Editor Comments (optional):

Reviewers' comments:

Reviewer's Responses to Questions

**Comments to the Author**

1. If the authors have adequately addressed your comments raised in a previous round of review and you feel that this manuscript is now acceptable for publication, you may indicate that here to bypass the “Comments to the Author” section, enter your conflict of interest statement in the “Confidential to Editor” section, and submit your "Accept" recommendation.

Reviewer #2: All comments have been addressed

2. Is the manuscript technically sound, and do the data support the conclusions?

Reviewer #2: Yes

3. Has the statistical analysis been performed appropriately and rigorously? 

Reviewer #2: Yes

4. Have the authors made all data underlying the findings in their manuscript fully available?

Reviewer #2: Yes

5. Is the manuscript presented in an intelligible fashion and written in standard English?

Reviewer #2: Yes

6. Review Comments to the Author

Reviewer #2: I think the authors have correctly addressed the various points raised during the article review process. It is, in my opinion, now acceptable for publication in PLOS ONE. However, I do have one final minor suggestion that may help in understanding the non-trivial concepts of "modules" vs. "modes".

Line 676: The difference between processes by modules or modes is not necessarily obvious to readers. The authors could add a small sentence that defines more clearly and explicitly what differentiates the two.

7. PLOS authors have the option to publish the peer review history of their article (what does this mean?). If published, this will include your full peer review and any attached files.

Reviewer #2: **Yes: **Sylvain Delplanque

---

## [Editor Report · Acceptance letter]

15 Jul 2020

PONE-D-19-24559R2 

A network model of affective odor perception 

Dear Dr. Toet:

I'm pleased to inform you that your manuscript has been deemed suitable for publication in PLOS ONE. Congratulations! Your manuscript is now with our production department. 

Kind regards, 

on behalf of

Dr. Alberto Greco 

Academic Editor

PLOS ONE